# The Influence of Organic and Mineral Fertilizers on the Quality of Soil Organic Matter and Glomalin Content

**Jiří Balík \*, Martin Kulhánek**  **, Jindřich Černý**  **, Ondřej Sedlář**  **, Pavel Suran and Dinkayehu Alamnie Asrade**

Department of Agro-Environmental Chemistry and Plant Nutrition, Faculty of Agrobiology, Food and Natural Resources, Czech University of Life Sciences, 165 00 Prague, Czech Republic; kulhanek@af.czu.cz (M.K.); cernyj@af.czu.cz (J.Č.); sedlar@af.czu.cz (O.S.); suranp@af.czu.cz (P.S.); asrade@af.czu.cz (D.A.A.)

\* Correspondence: balik@af.czu.cz

**Abstract:** The influence of different fertilizers (mineral/organic) on the quantity and quality of soil organic matter was monitored in long-term stationary experiments (27 years) with silage maize monoculture production on Luvisol. The main aim of this study was to investigate the relationship between easily extractable glomalin (EEG), total glomalin (TG), and parameters commonly used for the determination of soil organic matter quality, i.e., the content of humic acids ($C_{HA}$), fulvic acids ($C_{FA}$), and potential wettability index (PWI). A significant correlation was found between EEG content and $C_{SOM}$ content, humic acid content ($C_{HA}$), humic acid/fulvic acid ratio ($C_{HA}/C_{FA}$), PWI, and index of aromaticity (IAR). Furthermore, the contents of EEG and TG correlated with soil organic carbon ($C_{SOM}$). Periodical application of sewage sludge and cattle slurry increased the content of glomalin in soils. From the results, it is obvious that data about glomalin content can be used to study soil organic matter quality. A more sensitive method (a method that reacts more to changes in components of soil fertility) seems to be the determination of EEG rather than TG. The factors supporting use of EEG extraction in agronomic practice are mainly the substantially shorter time of analysis than TG, $C_{HA}$, and $C_{FA}$ determination and lower chemical consumption. Furthermore, the PWI method is even suitable for studying soil organic matter quality. On the other hand, the humus quality ratio (E4/E6) does not provide relevant information about soil organic matter quality.

**Keywords:** long-term experiments; Luvisol; cattle slurry; sewage sludge; humic substances fractionation; potential wettability index; glomalin

## 1. Introduction

Glomalin-related soil protein (GRSP) is produced on arbuscular mycorrhizal fungi (AMF) hyphal walls and it is a heat-stable glycoprotein. It can remain in the soil for years and is resistant to microbial attacks. Wright and Upadhyaya [1] used the terms easily extractable glomalin (EEG) and total glomalin (TG). However, it has been demonstrated that the fraction of soil organic matter (SOM) yielded under repeated autoclaving of soil in citrate buffer contains a mixture of various proteins as well as other compounds, such as humic acids [2,3]. The procedure used for TG extraction also probably does not extract all glomalin present in a soil [4]. Unlike TG, one extraction cycle is sufficient to obtain the EEG pool. The content of both the EEG and TG is proportional to increasing AMF colonization [5–7].

GRSP binds mineral and organic particles to soil aggregates for long periods, as it contains up to 85% of polysaccharides that are resistant to microbial decomposition [8] and acts as a sticky and insoluble biofilm that glues together minerals, clays, organic matter, and microorganisms [4,9,10].

Total glomalin contents in soil have tight linkages to total soil organic matter (SOM) contents [11–13]. On the other hand, there is also evidence of a negative correlation between GRSP and SOM content [14]. A strong positive correlation between glomalin

and soil fertility indicators (i.e., soil C, N, and P) has also been found [15–17]. High soil P content decreases AMF colonization [18] and the EEG content [19]. In contrast, the GRSP increases with the indicators of lower soil fertility, such as a high C:N ratio [19]. However, increasing GRSP content with increasing SOC (soil organic carbon), available P, total N, and K have also been observed [20].

Long-term application of poultry, pig, or cattle manure [21] and the application of compost [22–25] or farmyard manure [22,24–26] also increases GRSP content. Higher GRSP content was also reported after the application of sewage sludge [13,27] and straw [28,29]. A study by Nie et al. [28] further mentions that increased GRSP content in soil was found after the application of a mixture of straw and mineral fertilizer. Enhanced amounts of GRSP have also been observed after the application of different organic fertilizers, such as litter [22,24].

Many studies have demonstrated that humic acids, polyphenolic compounds, sugars, and lipids interfere with the Bradford assay [30]. The aforementioned organic compounds cross-react with the Bradford reagent, leading to the misinterpretation of the accurate GRSP content [31].

Generally, in the upper 0–10 cm soil depth, N and C contained in GRSP represent 5% of total soil N and 3.20% of total soil $C_{SOM}$ [19], where whole GRSP compounds are representing up to 25% of total $C_{SOM}$ in soil [15]. A positive correlation between GRSP and microbial biomass C and microbial respiration has been found in multiple soil environments [32,33]. A significant part of GRSP has long-term persistence (up to 42 years or more) in the soil medium, improving soil C sequestration [34].

GRSP production is seasonally dependent; AMF biomass peaks in the spring and summer, and consequently, GRSP content strongly increases in the summer [35]. Higher content of TG was also reported in the soil during the maize growth compared to the soil glomalin content before sowing [14].

Several recent publications have reviewed available information about the GRSP and its extraction, quantification, molecular characterization, and nomenclature in more detail [5,7,17,36,37]. Due to its positive correlation with soil organic matter carbon ($C_{SOM}$) content, the GRSP content is considered an indicator of changes in the quality of soil fertility [1,13,26,38,39].

This work aimed to monitor the possibility of using information about the GRSP content relative to soil organic matter quantity and even quality. Long-term experiments on Luvisol with silage maize and various mineral and organic fertilizers were used to study this issue. Maize was used in our experiment because it is one of the most produced crops across the world. The worldwide area used for maize production is about 16.8 million ha [40]. The Luvisols comprise over 500–600 million ha worldwide. Luvisols occupy mostly temperate regions, such as parts of the West Siberian Plain, East European Plains, Central Europe, the Mediterranean region, southern Australia, and the northern United States of America [41]. Luvisol represents 4.29% of the area of agriculturally used soils in the Czech Republic [42]. Most Luvisols are fertile soils and are suitable for a wide range of agricultural uses [41].

## 2. Materials and Methods

The study was carried out based on the long-term fertilization experiments since 1993 at the experimental stations located in Červený Újezd in the Czech Republic. Basic soil–climatic characteristics are given in Table 1.

**Table 1.** Basic description of the Červený Újezd experimental site at the start of the experiment.

| Location | Červený Újezd | References |
|---|---|---|
| GPS coordinates | 50°4′22″ N, 14°10′19″ E | |
| Altitude (m above sea level) | 410 | |
| Mean annual temperature (°C) | 7.70 | |
| Mean annual precipitation (mm) | 493 | |
| Soil type | Haplic Luvisol | NRCS USDA [43] [1] |
| Soil texture | Loam | NRCS USDA [43] [1] |
| Clay (%) (<0.002 mm) | 5.40 | |
| Silt (%) (0.002–0.05 mm) | 68.1 | |
| Sand (%) (0.05–2 mm) | 26.5 | |
| Bulk density (g cm$^{-3}$) | 1.50 | |
| $C_{SOM}$ (%) | 1.26 | CNS [2] |
| pH (0.01 mol/L CaCl$_2$) | 6.50 | ISO 10390 [44] |
| CEC (mmol$_{(+)}$ kg$^{-1}$) | 118 | |

[1] Natural Resource Conservation Service–United States Department of Agriculture. [2] CNS analyzer (Elementar Vario Macro, Elementar Analysensysteme, Hanau-Frankfurt am Main, Germany).

*2.1. Experimental Design*

The field experiment was conducted in a randomized complete block design with a plot area of 170 m$^2$. The experiments comprised four treatments: no fertilization control (Con), urea ammonium nitrate (N), cattle slurry (CS), and sewage sludge (SS). All treatments, including the control, were replicated four times. The maize hybrids used were "Malta" (1993–1996), "Torena" (1997 and 1998), "DK 254" (1999), "Compact" (2000), "Etendard" (2001–2003), "Rivaldo" (2004–2011), "RGT Indexx" (2012–2014), and "RGT Sixxtus" (2015–2019), and they were planted on each plot at a density of 80 thousand plants ha$^{-1}$. The maize was sown at the end of April/start of May, with 70 cm between plant rows. Since 1993, the mineral N fertilizers were applied before sowing every year in spring. Organic fertilizers sewage sludge (SS), precipitated with FeSO$_4$ and Al$_2$(SO$_4$)$_3$, and cattle slurry (CS), were applied at a different rate every year in autumn (October) and immediately incorporated into the soil with ploughing (25 cm depth). The complete fertilizing design is shown in Table 2. For the mineral nitrogen treatments, no other nutrients and liming were used from the beginning of the experiment. For the organic nitrogen treatments, the dosage of other nutrients depended on the content of nutrients in sewage sludge or cattle slurry. The dose of organic fertilizers was calculated according to the total nitrogen content determined with the Kjeldahl analysis (Table 3).

**Table 2.** Experimental design, applied organic and nitrogen fertilizers.

| Treatment | Avg. Fresh Weight (t ha$^{-1}$ Year$^{-1}$) | Avg. Dry Weight (t ha$^{-1}$ Year$^{-1}$) | N (kg N ha$^{-1}$ Year$^{-1}$) |
|---|---|---|---|
| Con | - | - | 0 |
| N | - | - | 120 |
| SS | 11.2 | 3.39 | 120 |
| CS | 49.3 | 2.14 | 120 |

Con: control; N: urea ammonium nitrate; SS: sewage sludge; CS: cattle slurry.

**Table 3.** The content of C, N, P, and K in organic fertilizers.

| Fertilizer | C | N | P | K | C/N |
|---|---|---|---|---|---|
| | | **(% Dry Weight)** | | | |
| SS | 25.8 | 3.52 | 2.42 | 0.46 | 7.32:1 |
| CS | 28.5 | 5.60 | 1.09 | 4.87 | 5.08:1 |

SS: 879 kg C ha$^{-1}$ year$^{-1}$; CS: 610 kg C ha$^{-1}$ year$^{-1}$; SS supplied 44% more carbon than CS.

Land leveling was not performed, to avoid overlapping treatments. The chemical protection during vegetation was performed only against weeds according to the actual situation. Pests and diseases were not eliminated due to their low appearance in the harvest plots.

Two rows of maize aboveground biomass (20 m$^2$ per plot) were harvested at silage maturity (roughly 65% biomass moisture content, BBCH 75 vegetation stage) and weight to obtain the aboveground biomass yield (BY). Dry BY was calculated based on the dry mass ratio in the subsamples.

*2.2. Soil Analysis*

Topsoil (depth of 30 cm) analyses were performed with air-dried soil samples ($\leq$2 mm) collected in September 2019, after the maize harvest.

**Soil organic carbon ($C_{SOM}$)** content in air-dried samples of soils, sewage sludge, and cattle slurry was determined using oxidation on the CNS Analyzer Elementar Vario Macro (Elementar Analysensysteme, Hanau-Frankfurt am Main, Germany).

**Fractionation of humic substances ($C_{HS}$)** was performed according to Kononova [45] to obtain the pyrophosphate extractable fraction, which represents the sum of the carbon in humic acids ($C_{HA}$) and fulvic acids ($C_{FA}$). In brief, $C_{HA}$ and $C_{FA}$ were extracted from a 5 g soil sample with a mixture of 0.10 mol L$^{-1}$ NaOH and 0.10 mol L$^{-1}$ Na$_4$P$_2$O$_7$ (1:20, $v/v$) solution. The carbon of humic substances $C_{HS}$ and $C_{HA}$ was determined using the oxidimetric titration method. The content of $C_{FA}$ was calculated as the difference between $C_{HS}$ and $C_{HA}$.

**The humus quality (E4/E6)** was analyzed according to the spectrophotometric method. The soil samples were extracted using sodium pyrophosphate (0.05 M Na$_4$P$_2$O$_7$) and measured by the absorbance ratio at 400 and 600 nm [46] (Lambda 25 UV/Vis (Perkin Elmer, Waltham, MA, USA).

**Extractable organic carbon** was determined using CaCl$_2$ and hot water extraction.

**For the 0.01 mol/L CaCl$_2$ extraction ($C_{DOC}$),** the extraction agent 0.01 mol L$^{-1}$ CaCl$_2$ was used (1:10, $w/v$) [47]. The $C_{DOC}$ content was determined in fresh soil samples by segmental flow-analysis using the infrared detection on a Skalarplus System (Skalar, Breda, The Netherlands).

**Hot water extraction ($C_{HWE}$)** was used to assess extractable soil organic carbon. Soil samples were dried at 40 °C and extracted with water (1:5, $w/v$). The suspension was boiled for one hour [48]. The $C_{HWE}$ was determined by a segmental flow analysis using the infrared detection on a Skalarplus System (Skalar, Breda, The Netherlands).

**The potential wettability index (PWI)** and **index of aromaticity (IAR)** were determined using DRIFT (diffuse reflectance infrared Fourier transform spectroscopy) spectra. DRIFT spectra were recorded by the infrared spectrometer (Nicolet IS10, Waltham, MA, USA). The spectra with a range of 2.50 to 25.0 μm (4000 to 400 cm$^{-1}$) were used. The gold mirror was used as a background reference. The 64 scans with a resolution of 4.00 cm$^{-1}$ and Kubelka–Munk units were applied. OMNIC 9.2.41 software (Thermo Fisher Scientific Inc., Waltham, MA, USA) was applied for spectra analysis. The bands of the alkyl C–H groups-A (2948–2920 cm$^{-1}$ and 2864–2849 cm$^{-1}$) were assumed to indicate the hydrophobicity, and bands of the C=O groups-B (1710 and 1640–1600 cm$^{-1}$) indicated hydrophilicity. The ratio of hydrophobicity and hydrophilicity was used to determine the potential wettability index [49].

$$PWI = A/B$$

The aromaticity index was calculated according to the reflectance of aliphatic bands ranging from 3000–2800 cm$^{-1}$ (AL) and aromatic band at 1520 cm$^{-1}$ (AR) [50].

$$IAR = AL/(AL + AR)$$

**Easily extractable glomalin (EEG)** and **total glomalin (TG)** were performed according to Wright and Upadhyaya [38], i.e., to 1.00 g of ground dry-sieved soil, 8 mL of sodium

acetate citrate (20 mmol $L^{-1}$ of pH 7.0-EEG, 50 mmol L pf pH 8.0-TG) was added, followed with autoclaving at 121 °C (30 min-EEG, 60 min-TG), cooling down and centrifugation at 5000 rpm (10 min-EEG, 15 min-TG). In the case of the TG, the centrifugation of the supernatant of the same sample was repeated 5 times until the supernatant no longer showed the red–brown color typical for glomalin.

**Mehlich 3** extraction was conducted according to Mehlich [51] to determine the available phosphorus content (**P$_{M3}$**). The 3.00 g soil samples were extracted with a 30 mL solution of 0.02 mol $L^{-1}$ $CH_3COOH$, 0.25 mol $L^{-1}$ $NH_4NO_3$, 0.015 mol $L^{-1}$ $NH_4F$, 0.013 mol $L^{-1}$ $HNO_3$, and 0.001 mol $L^{-1}$ ethylenediaminetetraacetic acid (EDTA). The solution was shaken for 5 min on a horizontal shaker and subsequently filtered. Phosphorus content in extracts was measured by the ICP-OES.

**The pH determination (pH$_{CaCl_2}$)** was conducted according to ISO 10390 [44] with slight modification. A total of 5.00 g of soil sample was added to 25 mL of $CaCl_2$ solution. The slurry was shaken for 60 min and was subsequently left to rest for 60 min. After this period, pH was measured using a WTW 330i meter (Xylem Analytics, Weilheim, Germany).

**Statistical analysis**

The results were assessed using ANOVA statistical analysis with Tukey's test using the Statistica program (TIBCO, Paolo Alto, CA, USA). Principal component analysis (PCA) was performed to evaluate the relationships between the content of glomalin (EEG, TG) and qualitative parameters of SOM using XLSTAT (Addinsoft, New York, NY, USA). The variables were submitted to PCA, and eigenvalues > 1, variance (%), and cumulative (%) criteria were used to define the association among the variables.

**List of variables**

For better transparency and orientation in the text, Table 4 shows a list of used variables abbreviations, which are commonly used in the following chapters, including the variables units or scale.

**Table 4.** Description of variables.

| Abbrev. | Full Description of Variables | Unit/Scale |
|---|---|---|
| $C_{SOM}$ | Soil organic carbon | % |
| EEG | Easily extractable glomalin | mg kg$^{-1}$ |
| T | Total glomalin | mg kg$^{-1}$ |
| $C_{HWE}$ | Carbon–hot water extraction | mg kg$^{-1}$ |
| $C_{DOC}$ | Carbon–0.01 M $L^{-1}$ $CaCl_2$ extraction | mg kg$^{-1}$ |
| $C_{HA}$ | Carbon in humic acids | mg kg$^{-1}$ |
| $C_{FA}$ | Carbon in fulvic acids | mg kg$^{-1}$ |
| $C_{HS}$ | Carbon humic substances | mg kg$^{-1}$ |
| $C_{HA/FA}$ | Ratio $C_{HA}$ and $C_{FA}$ | - |
| E4/E6 | Humus quality–ratio (E4/E6) | - |
| $P_{M3}$ | Mehlich-3 extractable phosphorus | mg P kg$^{-1}$ |
| $N_t$ | Total nitrogen | % |
| $C_{SOM}/N_t$ | Ratio $C_{SOM}$ and $N_t$ | - |
| $C_{DOC}/N_{DOC}$ | Ratio $C_{DOC}$ and $N_{DOC}$ | - |
| PWI | Potential wettability index | - |
| IAR | Aromaticity index | - |
| $pH_{CaCl_2}$ | Soil pH | - |

## 3. Results

The average yield of maize biomass on Con treatment during the entire experiment (1993–2019) was 9.00 t of DM ha$^{-1}$. Biomass yields on SS, N, and CS treatments increased by 35, 39, and 44%, respectively (Table 5). The influence of N fertilizer and different organic fertilizers is also reflected in soil organic matter content ($C_{SOM}$) (Table 5). The content of $C_{SOM}$ at the beginning of the experiment was 1.26%. After 27 years, the $C_{SOM}$ content on Con treatment amounted only up to 0.981%, which is a decrease of 22%. The lowest $C_{SOM}$ content was observed under mineral nitrogen fertilization (N treatment). It is generally

accepted that intensive and exclusive nitrogen fertilization increases the mineralization of soil organic matter. A significantly higher $C_{SOM}$ content was observed in the SS treatment, which was also greater than that in the CS treatment. This is also in agreement with the content of organic substances on SS in comparison with CS. SS treatment was annually supplied with 269 kg C ha$^{-1}$ more (44% more) than CS. A greater C:N ratio on SS (7.32:1) than CS (5.08:1) could have also influenced the intensity of mineralization of fertilizer-applied organic matter and soil organic matter. An important qualitative indicator of SOC is the content of humic acids (HA) and their ratio with fulvic acids (FA). In general, the quality of organic matter is higher with increasing content of HA and decreasing content of FA. The content of $C_{HA}$ was significantly higher after the application of SS. There was also an increasing tendency in $C_{HA}$ content on CS treatment; however, this increase was not significant in comparison with Con and N treatments. There were no significant differences in the content of $C_{FA}$, but there was an increasing trend in SS. This treatment also produces significant differences in $C_{HA}/C_{FA}$ ratio in comparison with Con in particular.

**Table 5.** Qualitative and qualitative parameters of soil organic matter (SOM); soil pH values and maize dry biomass yield (BY).

| Parameter/Treatment | Con | N | SS | CS |
|---|---|---|---|---|
| $C_{SOM}$ (%) | 0.98 [a] | 0.95 [a] | 1.20 [b] | 1.12 [ab] |
| EEG (mg kg$^{-1}$) | 578 [a] | 584 [a] | 670 [b] | 633 [ab] |
| TG (mg kg$^{-1}$) | 1950 [a] | 2130 [ab] | 2370 [b] | 2110 [ab] |
| $C_{HWE}$ (mg kg$^{-1}$) | 186 [a] | 172 [a] | 243 [b] | 240 [b] |
| $C_{DOC}$ (mg kg$^{-1}$) | 18.5 [a] | 22.5 [b] | 29.1 [c] | 23.0 [b] |
| $C_{HA}$ (mg kg$^{-1}$) | 0.078 [a] | 0.090 [ab] | 0.123 [b] | 0.110 [ab] |
| $C_{FA}$ (mg kg$^{-1}$) | 0.138 [a] | 0.145 [a] | 0.158 [a] | 0.133 [a] |
| $C_{HS}$ (mg kg$^{-1}$) | 0.215 [a] | 0.235 [a] | 0.280 [b] | 0.243 [ab] |
| $C_{HA/FA}$ | 0.565 [a] | 0.629 [ab] | 0.784 [b] | 0.836 [b] |
| E4/E6 | 3.53 [a] | 3.63 [a] | 3.65 [a] | 3.65 [a] |
| $P_{M3}$ (mg kg$^{-1}$) | 142 [a] | 102 [a] | 366 [c] | 304 [b] |
| $N_t$ (%) | 0.096 [a] | 0.102 [a] | 0.120 [b] | 0.111 [ab] |
| $C_{SOM}/N_t$ | 10.2 [b] | 9.31 [a] | 9.97 [b] | 10.1 [b] |
| $C_{DOC}/N_{DOC}$ | 0.550 [b] | 0.161 [a] | 1.03 [c] | 0.954 [c] |
| PWI | 0.010 [ab] | 0.010 [a] | 0.014 [b] | 0.014 [b] |
| IAR | 0.020 [a] | 0.019 [a] | 0.026 [b] | 0.027 [b] |
| $pH_{CaCl_2}$ | 6.36 [ab] | 5.91 [a] | 6.25 [ab] | 6.70 [b] |
| Maize BY (t ha$^{-1}$) | 9.00 [a] | 12.5 [b] | 12.2 [b] | 13.0 [b] |

Different letters describe statistically significant differences between treatments. Tukey's LSD test; $p < 0.05$.

Changes in quantity and quality of soil organic matter are also visibly reflected in the content of carbon extractable with hot water ($C_{HWE}$) and with weak extractant 0.01 M L$^{-1}$ $CaCl_2$ ($C_{DOC}$). The content of $C_{HWE}$ on both organic treatments was significantly higher than all other treatments, while the content of $C_{DOC}$ was significantly higher only in comparison with Con. The soil organic matter quality was also investigated using the E4/E6 method with sodium pyrophosphate (0.05 mol L$^{-1}$ $Na_4P_2O_7$) as an extractant. This method did not produce any significant differences among treatments (Table 5). The question is whether the E4/E6 method is quite suitable for evaluating soil organic matter quality.

The mineralization rate of soil organic matter is substantially influenced by the intensity and form (organic/mineral) of nitrogen fertilization. This is the reason for the inclusion of $N_t$ and $C_{SOM}/N_t$ ratio as well as 0.01 mol L$^{-1}$ $CaCl_2$ ($C_{DOC}/N_{DOC}$) (Table 5). It is also necessary to mention that nitrogen uptake by aboveground biomass was substantially higher than applied nitrogen (120 kg N ha$^{-1}$).

The content of $N_t$ was lowest in the non-fertilized Con treatment, which is following our supposition. On the other hand, the highest $N_t$ content was observed during the

SS treatment. Considering the stability of SOM, the C/N is more important. This ratio was highest in the non-fertilized Con treatment. It is possible that SOM is in the most stable configuration against mineralization in the Con treatment. Mineral fertilization (N treatment) displayed a significantly lower $C_{SOM}/N_t$ ratio (9.31:1). Differences in the $C_{DOC}/N_{DOC}$ ratio are even more visible with organic fertilizers compared to Con and N treatments than $C_{SOM}/N_t$.

For the evaluation of soil organic matter quality, a DRIFTS method was also used. Usually, this method is used to study bands of hydrophobic alkyl C–H groups and hydrophilic C=O groups of soil organic matter at 2948–2920 cm$^{-1}$ and 1710, 1640–1600 cm$^{-1}$, respectively. The ratio of hydrophobic and hydrophilic groups determines the potential wettability index (PWI). Ordinarily, PWI is directly proportional to the stability of soil aggregates and the quality of soil organic matter. It is clear that the lowest values were measured in the N treatment, while the highest values were measured in the organic fertilizer treatments (Table 5). Another studied criterion using the DRIFTS method was the aromaticity index (IAR) estimation. This index is calculated based on the reflectance of aliphatic compounds (3000–2800 cm$^{-1}$) and aromatic compounds (1520 cm$^{-1}$). Evidently, the usage of organic fertilizers increased the values of IAR. The lowest values were again produced by the N treatment.

Concerning the study of glomalin, the content of plant-available phosphorus using the extraction method Mehlich 3 was also included in the monitored parameters. In our experiments, certain treatments were annually supplied with phosphorus at rates of 82.1 kg P ha$^{-1}$ in SS and 23.2 kg P ha$^{-1}$ in CS. The SS treatment was significantly positive in the balance of incoming and outcoming (only plant uptake) phosphorus. On the other hand, the CS treatment had a slight, significantly negative balance while other treatments (Con and N) had a distinctly negative balance. This was subsequently manifested in the content of available phosphorus in soil. Treatments with organic fertilizers had significantly higher contents. From the values of the correlation coefficients (Table 6), it is clear that there was a significant positive correlation with EEG and an insignificant correlation with TG.

**Table 6.** Relationship between glomalin and other soil parameters.

| | $C_{SOM}$ | $C_{HWE}$ | $C_{DOC}$ | $C_{HA}$ | $C_{FA}$ | $C_{HS}$ | $C_{HA/FA}$ | E4/E6 |
|---|---|---|---|---|---|---|---|---|
| EEG | 0.527 * | 0.424 | 0.393 | 0.590 * | 0.004 | 0.454 | 0.587 * | −0.157 |
| TG | 0.521 * | 0.440 | 0.669 ** | 0.558 * | 0.349 | 0.609 * | 0.405 | 0.420 |
| $C_{SOM}$ | | 0.872 *** | 0.654 ** | 0.706 ** | 0.241 | 0.666 ** | 0.583 * | −0.131 |
| | $N_t$ | $P_{M3}$ | $C_{SOM}/N_t$ | $C_{DOC}/N_{DOC}$ | PWI | IAR | $pH_{CaCl_2}$ | |
| EEG | 0.472 | 0.612 * | 0.244 | 0.399 | 0.735 ** | 0.688 ** | 0.505 * | |
| TG | 0.592 * | 0.466 | −0.100 | 0.394 | 0.333 | 0.335 | −0.196 | |
| $C_{SOM}$ | 0.938 *** | 0.839 *** | 0.356 | 0.752 *** | 0.799 *** | 0.797 *** | 0.530 * | |

Pearson correlation coefficient: * $p < 0.05$; ** $p < 0.01$; *** $p < 0.001$.

Based on the presented results, it is clear that this is a very diverse set of criteria from the perspective of soil organic matter quality as well as from the perspective of nutrient content (nitrogen, phosphorus). This set of criteria was consequently used for testing glomalin as an indicator of soil organic matter quality. Values of easily extractable glomalin content (EEG) showed an increase in the organic fertilizer treatment. Total glomalin content (TG) was significantly higher in the SS treatment in comparison with the control. Correlation coefficient values comparing EEG, TG, and qualitative components of SOM are presented in Table 6.

The different variables were evaluated using principal component analysis (PCA). The principal components and their association were systematically selected based on three criteria, i.e., eigenvalue > 1.1, loading factors > 0.70, and percentage of variability > 6.50%, as mentioned in Table 7. The biplot position of variables explains 93.5% (PC1 and PC2) of the total cumulative variance, where the first factor describes 72.1%, the second 21.4%, and

the third 6.50% (Figure 1). PC1, which explained 72.1% of the total variance, was highly dominated by positively associated variables such as $P_{M3}$, EEG, IAR, $N_t$, PWI, $C_{HWE}$, $C_{HS}$, $C_{HA}$, $C_{HA/FA}$, $C_{DOC}/N_{DOC}$, E4/E6, $C_{SOM}$, $C_{DOC}$, and TG. The second PC2 explains about 21.4% of the variance and was positively correlated with $C_{SOM}/N_t$ and $pH_{CaCl_2}$ and negatively associated with $C_{FA}$. The third PC explains 6.5% of the total variance, and no variables dominated within the principal component. The $P_{M3,}$ EEG, $C_{SOM}$, and variables are strongly associated, and the most significant variables dominated within the PC1. $C_{SOM}$ was positively correlated with $N_t$ ($p < 0.05$; r = 0.94), $C_{HA}$ ($p < 0.05$; r = 0.71), PWI ($p < 0.05$; r = 0.80), $C_{HWE}$ ($p < 0.05$; r = 0.87), and IAR ($p < 0.05$; r = 0.80).

**Table 7.** The principal components (PCs) or factors and their loading factor values, eigenvalues, and variabilities (%).

| Parameters | PC1 | PC2 | PC3 |
|---|---|---|---|
| $C_{SOM}$ | 0.981 | 0.074 | −0.179 |
| EEG | 0.972 | 0.147 | 0.185 |
| TG | 0.800 | −0.590 | −0.107 |
| $C_{HWE}$ | 0.960 | 0.278 | −0.028 |
| $C_{DOC}$ | 0.855 | −0.509 | −0.101 |
| $C_{HA}$ | 0.980 | −0.183 | 0.073 |
| $C_{FA}$ | 0.397 | −0.778 | −0.488 |
| $C_{HS}$ | 0.884 | −0.446 | −0.141 |
| $C_{HA/FA}$ | 0.925 | 0.129 | 0.356 |
| E4/E6 | 0.760 | −0.378 | 0.528 |
| $P_{M3}$ | 0.980 | 0.141 | −0.139 |
| $N_t$ | 0.972 | −0.237 | −0.001 |
| $C_{SOM}/N_t$ | 0.259 | 0.822 | −0.508 |
| $C_{DOC}/N_{DOC}$ | 0.883 | 0.41 | −0.231 |
| PWI | 0.943 | 0.323 | 0.077 |
| IAR | 0.946 | 0.316 | 0.071 |
| $pH_{CaCl_2}$ | 0.435 | 0.895 | 0.098 |
| Eigenvalue | 12.3 | 3.60 | 1.10 |
| Variability (%) | 72.1 | 21.4 | 6.50 |
| Cumulative variance | 72.1 | 93.5 | 100 |

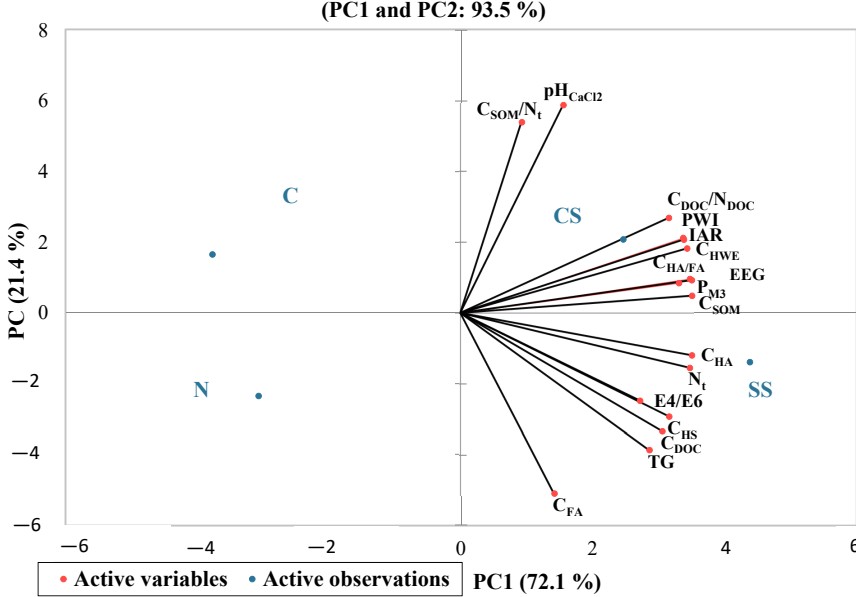

**Figure 1.** The biplot position of variables is determined by the first two principal components (PC1 vs. PC2); the red and blue dots represent active variables and observations, respectively.

The application of sewage sludge and cattle slurry to soil positively influenced the tested variables.

## 4. Discussion

The global trend of moving away from mineral fertilizers and returning to organic fertilizers creates a necessity for research of alternative and potentially available sources. Application of sewage sludge on arable soil is one of these alternatives. Sewage sludge is organic compounds and nutrients (P in particular), but it also may be a source of potentially toxic elements and substances. Therefore, extraordinary attention must be paid to their use. Production of sewage sludge is increasing in proportion to the increasing population worldwide. About 37% of produced sludge is concurrently used as fertilizer in EU countries [52]. Due to the aforementioned facts, the results of this study, focusing on soil carbon transformations, could be helpful on a greater than regional scale.

One of the benefits of our experiments is a small set of variable factors that could influence changes in the quality and quantity of soil organic matter. Long-term silage maize production can be considered a deciding factor. It is generally accepted that maize belongs to a group of crops substantially dependent on arbuscular mycorrhiza and, at the same time, promotes its development [53]. Another variable was differences in the quality of applied fertilizers (mineral/organic). Nevertheless, all fertilized treatments received 120 kg N ha$^{-1}$ year$^{-1}$. The long duration of experiments (27 years) showed its influence on the quantity and quality of soil organic matter (Table 5). Therefore, it can be assumed that this set of different qualitative parameters was suited for monitoring changes in glomalin content in soils. It is also necessary to remember that experiments had classic tillage treatment with roughly 25 cm ploughing depth.

The following methods were used to determine the changes in soil organic matter quality and quantity: $C_{SOM}$, hot water extraction, 0.01 mol L CaCl$_2$ extraction, $N_t$ assessment, and alternative determination of $C_{SOM}/N_t$ and $C_{DOC}/N_{DOC}$ ratios. These methods are relatively sensitive and reproducible [48,54]. Fractionation of humic substances, humic acids, and fulvic acids [45] is a very time-consuming process, with a wide variety of possible analytical inaccuracies that severely limit this method in terms of extensive soil monitoring. If performed correctly, this method is capable of producing very good results characterizing soil organic matter. Our study's results agree with Thai et al. [55] and demonstrate that the E4/E6 method is unfit for the determination of soil organic matter quality, and was therefore not used in further evaluations.

### 4.1. Content of Glomalin

GRSP exists in two forms: easily extractable form (EEG) and total form (TG). The determination of EEG is based on a single extraction using sodium citrate. TG is repeatedly extracted several times. On the other hand, EEG is easily extractable with relatively high solubility in water in comparison with TG [38]. The average contents of TG and EEG were 2140 mg kg$^{-1}$ and 616 mg kg$^{-1}$, respectively. EEG, therefore, makes up to 29% of TG. This ratio is greater than that published in an earlier study (18%) on another experiment with crop rotation [13].

Total glomalin content (TG) in our experiments was between 1950 mg kg$^{-1}$ and 2370 mg kg$^{-1}$ and is in accord with the results of Agnihotri et al. [56], who state that glomalin content in regular soils lies within the interval of 2.00–15.0 mg g$^{-1}$. Lower TG values between 1.00–9.00 mg g$^{-1}$ are presented in other works [14,27,56–59].

Glomalin can add up to 25% of the total $C_{SOM}$ content of soil organic matter [15]. This proportion lies in the range of 18.8 to 22.4% in our experiments based on treatment and is in good agreement with the aforementioned authors. Glomalin contains roughly 28–48% of carbon [60]. The mean value of this interval is 36%, and using this value, we calculated carbon content in glomalin ($C_{gl}$) in our experiments based on treatment to be Con: 702 mg $C_{gl}$ kg$^{-1}$, N: 767 mg $C_{gl}$ kg$^{-1}$, SS: 853 mg $C_{gl}$ kg$^{-1}$, and CS: 760 mg $C_{gl}$ kg$^{-1}$. Content of $C_{SOM}$ in the Con treatment was determined at 0.981% (9810 mg C kg$^{-1}$). Using

further calculations, we find that proportion of glomalin carbon ($C_{gl}$) is 7.2% in $C_{SOM}$ in the Con treatment. On other treatments, this proportion is N: 8.1%, SS: 7.1%, and CS: 6.9%. These results are distinctly higher than the 3.2% that was published in Lovelock et al. [19]. If 28% were to be used (instead of 36%), the resulting proportion of $C_{gl}$ in $C_{SOM}$ would be greater than 5.2%. Using a calculation with 36% of carbon in glomalin for easily extractable glomalin (EEG), the resulting proportion of $C_{EE}$ in $C_{SOM}$ would be 1.9–2.2%.

Nitrogen content in glomalin shows a relatively wide interval of 0.9–7.3% [61,62]. Treseder and Turner [63] present a substantially tighter interval of 2–4%. Using calculation with 3% of nitrogen content, the resulting nitrogen content in glomalin ($N_{gl}$) in soil based on treatment is Con: 59.0 mg $N_{gl}$ kg$^{-1}$, N: 64.0 mg $N_{gl}$ kg$^{-1}$, SS: 71.0 mg $N_{gl}$ kg$^{-1}$, and CS: 63.0 mg $N_{gl}$ kg$^{-1}$. Therefore, the proportion of $N_{gl}$ in $N_t$ is between 5.70% and 6.30% based on the treatment. These calculated values are in very good agreement with the data of Lovelock et al. [19], who presented a 5% portion of $N_{gl}$ in $N_t$.

*4.2. Relationship between Glomalin and SOM Parameters*

Organic fertilization increases the proportion of macroaggregates in soil and the average size of these aggregates parallel with an increase of microbial biomass and the contents of $C_{SOM}$ and GRSP [22]. The application of SS especially increased the contents of $C_{SOM}$ and GRSP (Table 5). The relationship between $C_{SOM}$ content, as well as TG and EEG content, is further confirmed by correlation coefficient values (Table 6). A similar positive correlation is also presented in Singh et al. [15] and Wang et al. [16]. In the conditions of the Czech Republic, the increase of GRSP content with an increase of organic matter content was also presented by Šarapatka et al. [20]. In our experiment, the strength of the relationship between $C_{SOM}$, TG, and EEG is almost the same. The conclusions of Řezáčová et al. [23] were not confirmed, as they presented a tighter relationship between EEG and $C_{SOM}$ than TG and $C_{SOM}$.

In accordance with our results, an increase in glomalin content after the application of SS is also presented by Sandeep et al. [27] and Balík et al. [13]. Furthermore, there was no evidence suggesting that N fertilization increased glomalin content (Table 5), which conflicts with the results of previous studies [13,58,63]. The content of both GRSP forms (TG and EEG) is also in good correlation with more stable fractions of soil organic matter ($C_{SOM}$), that is, with the content of humic acids as opposed to the content of fulvic acids (Table 6). This indirectly confirms the hypothesis that glomalin is relatively resistant to mineralization. The persistence time of glomalin is up to 42 years, and the residence time is longer than the original organic matter in bulk soil (10–37 years) [34,64,65].

The generally accepted conclusion is that the $C_{SOM}/N_t$ ratio characterizes the potential for mineralization of soil organic matter in a certain way. N fertilization tightens this ratio [13], which was subsequently documented in N treatment in this study. Therefore, we were interested in the relationship between the $N_t$ content, $C_{SOM}/N_t$ ratio, and $C_{DOC}/N_{DOC}$ ratio on one side, and glomalin content on the other side. No significant relationship was found (Table 6). This could point towards greater stability of glomalin against mineralization than other soil organic matter. Jha et al. [66] found a positive correlation between the mineralization process (release of nitrogen from organic bonds) and EEG content. Authors further concluded that EEG content could be a certain criterion for estimating potential nitrogen mineralization from organic bonds. In our experiments, we have not confirmed this conclusion. Higher values of the correlation coefficient between TG and $N_t$ can be caused by the fact that during repeated extraction with sodium citrate, a portion of other organic compounds, including proteins, can be released aside from glomalin [4,5].

A high content of mobile phosphorus in soil reduces colonization of roots with AMF [17] and thereby also reduces the content of EEG in soil [18]. This was not proven in our experiments, but on the other hand, it does confirm the results of Šarapatka et al. [20], who stated an increase in GRSP with an increase of available phosphorus. Cissé et al. [67] did not observe changes in GRSP content during long-term maize production and periodic

phosphorus fertilization on sandy soil. The authors further mention that phosphorus fertilizer treatment had a slight trend toward increasing maize yield. Singh et al. [15] and Wang et al. [16] also observed a positive correlation between GRSP content and components of soil fertility, including phosphorus content. The disproportion of our results compared to Lovelock et al. [19] can be explained in the fact that even Con and N treatments had a good supply of available phosphorus. Maize plants were well supplied by phosphorus on all treatments, meaning there were no conditions for increased stimulation of AMF growth.

Greater content of GRSP was usually found on more acidic soils in comparison with neutral or carbonate soils. Glomalin content increased with the reduction of pH value [56]. In our experiments, an opposite effect was observed. A significant positive correlation was found between EEG content and $pH/_{CaCl_2}$ solution. Apparently, soil reaction did not have a dominant influence over glomalin content. Other factors had a decisive influence (e.g., $C_{SOM}$). It is important to note the influence of nutrients in organic fertilizers, whose presence leads to a significant increase in plant biomass, including root development, and, as a result, also increases glomalin production. Higher soil organic matter content also leads to the formation of bigger soil aggregates, inside which a bigger glomalin content is bound. This also reduces the intensity of mineralization.

*4.3. Potential Wettability Index*

The quality of soil organic matter can be described by the ratio of aliphatic (C–H) and carboxyl (C=O) bonds called the potential wettability index (PWI) [68,69]. PWI can be used (a) during the determination of stability of soil aggregates, (b) for water sorption properties of soil, and (c) for the description of soil organic matter quality. Values of PWI were in the interval of 0.010–0.014 (Table 5). Fér et al. [70] mention values of PWI for Luvisol type between 0.010–0.030, which agrees with our values. It is obvious that organic fertilization significantly increased values of PWI in comparison with the Con and N treatments. Organic fertilization, therefore, increased the fraction of hydrophobic particles and contributed to the formation of bigger soil aggregates. Adani [71] also measured an increase in aliphatic carbon fraction in humic acids after application of SS after 10 years of experiments with a dose of 1 t DM $ha^{-1}$ $year^{-1}$. Demyan et al. [72] observed an increase in value after the application of manure. Leue et al. [73] found a positive correlation between SOM content and PWI values. Values of PWI in our experiments also showed a very strong and significant correlation with $C_{SOM}$ content. GRSPs are temperature-stable, sticky, and hydrophobic glycoproteins [38]. Following a simple though-chain, the following conclusion can be deduced: with an increase in glycoproteins content, there is also an increase in hydrophobic particle content and an increase of PWI. A significant correlation between EEG and PWI (0.735, $p < 0.01$) is in accordance with this hypothesis. The aromaticity index (IAR) also expresses the stability of soil organic matter [55]. Treatments with organic fertilizers produce significantly higher values. Similar to PWI, IAR also correlates with EEG.

**5. Conclusions**

The influence of different fertilizers (mineral/organic) on the quantity and quality of soil organic matter was monitored in long-term stationary experiments with silage maize monoculture production. Two of the observed factors were the contents of easily extractable glomalin (EEG) and total glomalin (TG). The contents of EEG and TG correlated with $C_{SOM}$. Periodical application of sewage sludge and cattle slurry increased the content of glomalin in soils. Data about glomalin content can be used to study soil organic matter quality; however, this parameter is not singular or exceptional. A more sensitive method (a method that reacts more to changes in components of soil fertility) seems to be the determination of EEG, rather than TG, in our experiments. A significant correlation was found between the EEG content and $C_{SOM}$ content, humic acid content ($C_{HA}$), $C_{HA}/C_{FA}$ ratio, potential wettability index (PWI), and aromaticity index (IAR). Another factor supports the greater practical application of this method, and that is a substantially shorter time of analysis

than TG determination. The potential wettability index method is suitable for studying soil organic matter quality. The E4/E6 method does not produce relevant soil organic matter quality results.

**Author Contributions:** Conceptualization, J.B. and M.K.; formal analysis, J.Č.; methodology, J.Č. and O.S.; writing: original draft, J.B.; laboratory and statistical analysis, P.S. and D.A.A. All authors have read and agreed to the published version of the manuscript.

**Funding:** This work was funded by the Ministry of Agriculture of the Czech Republic (Project No. QK21010124).

**Institutional Review Board Statement:** Not applicable.

**Informed Consent Statement:** Not applicable.

**Data Availability Statement:** Data available from corresponding author.

**Acknowledgments:** Thanks to the team of the Department of Agro-Environmental Chemistry and Plant Nutrition of the Czech University of Life Sciences in Prague for help and support, especially Jana Najmanová for help with laboratory work and measurements.

**Conflicts of Interest:** The authors declare no conflict of interest.

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
