# Peer review of "The Influence of Organic and Mineral Fertilizers on the Quality of Soil Organic Matter and Glomalin Content"

_agronomy, doi:10.3390/agronomy12061375_

Round 1
Reviewer 1 Report
The work is interesting and deals with an important issue taking into account the importance of carbon sequestration in soil. Cultivation in monoculture and unbalanced, one-sided nitrogen fertilization is not conducive to this, which has been confirmed. However, the importance of using natural / organic fertilizers began to be appreciated again. Unfortunately, in many countries municipal sewage sludge is thermally transformed in order to eliminate their storage prior to natural use. It is questionable whether slurry should be applied in the autumn, when a spring crop is being grown. The applied variants differ in many features that were not considered in these studies, not only in fertilization. Therefore results should be supplemented with the P content (or K because high maize requirements) in the materials used as fertilizers and in soil, and the yields of maize (or references to previous papers), because their effect on the content and quality of glomalins and organic matter is discussed (e.g. see page 12). In the absence of data, some statements cannot be verified, e.g. soil aggregates formation. The total content of glomalins is not a fraction, it is certainly easily extractable glomalins. Concentration it should be applied to solutions, gases or excessive amounts of xenobiotics rather than to the solid phase of the soil which is organic matter. Literature for the analytical methods used should be completed. The abbreviations (EE - EEG, T - TG) used and the rounding of numbers should be standardized throughout the paper. The manuscript should be read carefully to remove any deficiencies, e.g. lines 59, 75, 186, 275-276 (second time the same sentence), 300 (insert reference to Table 5), 320-322 and 432-442 (transfer part to introduction and results, respectively), 330 (instead too radical "also a source" insert "also may be a source"), 373. Conclusions are correct.
Author Response
Dear reviewer 1,
For better transparency, we prepared the PDF file with answers to all your comments and suggestions. Please see the attached file.
Yours sincerely
Prof. Ing. JiÅ™í Balík, CSc., dr.h.c.

Reviewer 2 Report
This is a very interesting and novel study with regards to the relationship between the GRSP content and soil organic matter quality. The Long-term experiments provide valuable information about the influence of mineral and organic fertilization on the soil organic matter parameters with a particular emphasis on the relationship between easily extractable glomalin and total glomalin and soil organic matter parameters. The whole manuscript is very well written and presented to the reader. The "Introduction" is based on recent findings and “Materials and Methods” are described with sufficient details. The results have been appropriately verified by statistical analysis. Tables and figures are very clear to the reader and understandable. Main results and ideas are well documented, justified and supported by relative references. Based on the above this article is strongly recommended to be published in the journal
Author Response
Dear reviewer #2,
Thank you very much for the positive review and for your warm words about our manuscript. We are glad that you agree with the importance of determination of organic matter quality parameters. It supports our thoughts that we are moving in good direction.
In the name of coauthors
Prof. Ing. JiÅ™í Balík, CSc., dr.h.c.
Reviewer 3 Report
Introduction:
Lines 29 – 36 seem rather methodological descriptions.
Line 59 reference 29, should be checked.
Materials and methods
What were the doses of the organic fertilizations? 170 kg ha-1?
Methods are missing from extractable organic carbon determinations line 158, and line 163.
Results
Line 213 – it should write down how the yields were determined in the material and methods chapter, or at least should be mentioned that the yields were investigated at all.
Table 6. might be removed – line 310
Figure 1. PC1 and 2: 93.5% can goes in the text, line 288.
Table 7. in line 315, unnecessary or can go to the Materials and methods chapter
Discussion
Probably the most problematic part of the manuscript. Now in this present form is kind of mixture of the introduction chapter and the result chapter, repeating the references and the results instead of exploring the significance of the work.
Line 373 references 4010?,23
Author Response

(The authors gave the same response as above.)

Reviewer 4 Report
Title:
I think the title needs to be corrected. First, the term 'fertilization' does not necessarily refer to types of 'fertilizers'. Rather it means 'the action or process' of fertilizing/applying fertilizers'. Since the experiment examined the influence of different fertilizers (organic and mineral), this should reflect in the title, for clarity.
Abstract:
The overall Abstract needs to be carefully written again with special focus on the objective of the study. What exactly is the novelty in this paper? Is the effects of organic amendments in increasing content of glomalin in soils special?
L16-17. How did the authors use (data on) glomalin content to study SOM quality, since quality of glomalin was not mentioned in this section. What was DRIFT used to measure, the quality of glomalin or SOM? No information on SOM quality was provided in this abstract. In L23-24, the use of PWI as a biomarker for studying SOM quality was given. But it is not clear how SOM quality actually related to PWI in the study. Also, there is no description on the type and effects of the mineral fertilizers used in comparison to the organics.
Introduction:
If GRSP is produced by AMF, did the authors determine AMF presence in the soil used in this experiment to determine AMF contribution to the production of GRSP? Did the amendments contribute to/influence AMF production of GRSP in this study?
The overall flow of this section can be better improved. Several sections can be reduced to prevent repetitions, particularly in the impact of organic fertilizers on GRSP (L43-65).
L70-71 is not clear. Does GRSP constitute 3.2% or 25% of total soil C? It cannot be both.
L71-72 is not clear. Given the microbial biomass C indicates immobilization (uptake) of C into microbial biomass, while microbial respiration indicates efflux (release) of C, it is not clear how GRSP can contribute positively to both?
L85-86. The application of GRSP content as a biomarker for SOM quality (not quantity) in this paper is not clear. Soil organic matter is usually a marker for soil organic carbon. Is it possible that the protein polymer is (completely) mineralized, but not the organic carbon, leading the high C:N ratio in the soil?
Author Response
Dear reviewer 4,
For better transparency, we prepared the PDF file with answers to all your comments and suggestions. Please see the attached file.
Yours sincerely
Prof. Ing. JiÅ™í Balík, CSc., dr.h.c.
